# SmGNN: Link Prediction in Sparse Layers of Multi-layer Graphs

## Abstract

Link prediction is a crucial task in multi-layer graphs for different applications, where real-world graphs often consist of multiple types of relations represented as different layers. However, these multi-layer graphs often suffer from missing edges, especially in specific layers with a high number of missing edges (sparse layers) due to privacy concerns. In this paper, we tackle the challenge of predicting missing links in such layers to enhance the link prediction performance in multi-layer graphs. Training a Graph Neural Network (GNN) directly for link prediction on the sparse layer with limited edges would be challenging for exploring missing links and may lead to sub-optimal performance. To tackle this problem, we propose a novel framework called Sparse Layer Reconstruction Multi-layer Graph Neural Network (SmGNN). SmGNN proposes to leverage information from other relation types (layers) to explore missing links in the sparse layer. By selectively fusing relevant information from other layers, we learn relevant representations that capture the characteristics of the sparse layer. Additionally, we incorporate node similarity information based on the relevant representation to enhance the graph structure of the sparse layer. By augmenting the graph structure, our approach improves the representation learning process and enables a more comprehensive exploration of relational patterns and connections within the sparse layer. Experimental evaluations on three real-world datasets demonstrate the effectiveness of our proposed SmGNN approach.

## 1 Introduction

Graphs play an essential role in many applications, including social networks (Qu et al., 2021), recommendation systems (Fan et al., 2019; Yu et al., 2021; Chang et al., 2021), and knowledge graphs (Nickel et al., 2015). As real-world graphs are often incomplete, link prediction (Zhou, 2021), which aims to predict missing links, is a critical task. For instance, link prediction can help new friends recommendation on social media (Adamic & Adar, 2003; Tan et al., 2019; Sankar et al., 2021), protein interaction prediction (Qi et al., 2006), and knowledge graph completion (Nickel et al., 2015; Nathani et al., 2019; Dong et al., 2023). Existing link prediction methods can be broadly classified into two categories: heuristic-based approaches (Lü et al., 2009; Newman, 2001; Adamic & Adar, 2003; Katz, 1953) and representation learning-based approaches (Acar et al., 2009; Kipf & Welling, 2016b; Zhang & Chen, 2018; Pan et al., 2022). Recently, Graph Neural Networks (GNNs) have emerged as a powerful tool for link prediction due to their great ability in node representation learning that captures both node attributes and local topology information (Kipf & Welling, 2016b; Zhang & Chen, 2018; Cai et al., 2021).

Despite the great success of GNNs for link prediction, most existing works focus on single-layer graphs, i.e., there is only one edge/relationship between a pair of nodes; while in real-world, there could be multiple edges/relationships between a pair of nodes, which can be described by *multi-layer graph*. Generally, a multi-layer graph has multiple layers with each layer containing connections in terms of one kind of relationship. Figure 1 shows an example of a 4-layer criminal network, which captures 4 different types of interactions and relationships among individuals involved in criminal activities, where Layer A denotes "likes" on social media, Layer B captures phone call interactions, Layer C means text message interactions and Layer D represents private meetings.

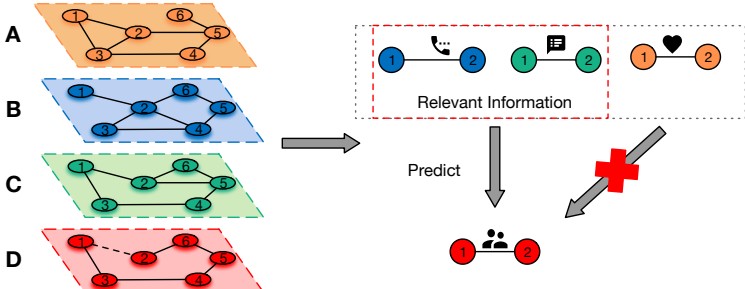

Figure 1: A motivation example of using other layers' information to reconstruct unobserved links in the sparse layer.

For many applications, the collected multi-layer graphs might have some dense layers that have more complete edge connections and some *sparse* layers that only have limited edge connections. And it is usually of great interest to predict the missing links in the sparse layer. For example, in criminal networks, some kinds of relationships can be easily tracked, such as phone calls, text messages, or e-mails, giving us dense layers; while some types of edges are very difficult to track, such as private meetings and co-crime activities (Bahulkar et al., 2018), resulting in sparse layers. It is very important to identify/predict these hidden/missing edges in sparse layers, e.g., private meetings and co-crime activities, which can help with criminal role identification, criminal network disruption, and future criminal prediction. However, due to the lack of edges for message passing of GNNs and for providing supervision to train the model, it is very challenging to directly apply existing link prediction methods, especially GNNs, to predict missing links for the sparse layer.

Though it is challenging to directly use the sparse layer for link prediction, the sparse layer might have correlations with other dense layers, which paves us a way to predict the missing links. For example, in Figure 1, Layer D is the sparse layer, which has a significant number of missing edges due to the covert nature of private meetings. In this network, if two nodes $v_1$ and $v_2$ have frequent phone call interactions (Layer B) and text message exchanges (Layer C), it provides an indication of a potential relationship that might also involve private meetings (Layer D). However, the "likes" relation may not be relevant to private meetings because it just indicates user engagement and interest in a particular post or content, and doesn't necessarily provide information about physical interactions or meetings. Therefore, to predict missing edges in Layer D (private meetings), we need to selectively extract relevant information from Layers B and C, excluding Layer A. This enables us to infer covert relationships through private meetings by leveraging the relevant information between Layers B and C. Though promising, the work on leveraging other layers' information to predict links for sparse layers in multi-layer graphs is rather limited.

Therefore, in this paper, we investigate a novel problem of link prediction for sparse layers in multi-layer graphs by leveraging link information from other layers. In essence, there are two main challenges: (**i**) how to learn useful information from other layers that can be relevant to the structural characteristics of the sparse layer; and (**ii**) how to utilize the acquired information from other layers for link prediction on the sparse layer with a limited number of edges. To resolve these challenges, we introduce a novel framework called Sparse Layer Reconstruction Multi-layer Graph Neural Network (SmGNN). We propose a relevant information encoder module that incorporates other layers' information into a learnable weighted fusion part to obtain relevant representations. Furthermore, we utilize supervised edge prediction signals on both the sparse layers and dense layers to assign varying weights to the dense layers, enabling the extraction of pertinent structural information that is specifically suited to the characteristics of the sparse layer. Then, we propose to employ a GNN model to encode the sparse layer and learn its representation through edge prediction. To enhance the availability of supervised signals, we augment the edge set on the sparse layer with additional edges derived from relevant representations from other layers. This augmentation facilitates the effective utilization of the augmented information, leading to improved encoding and prediction. Moreover, to avoid disregarding important relational patterns and connections, we incorporate the relevant representation while considering the graph structure. By incorporating the augmented graph structure with the original structure, we learn a more expressive representation that captures underlying patterns and relationships in the sparse layer more effectively. In summary, our main contributions are:

- We investigate a new problem of predicting missing links for a specific relation within multi-layer graphs, where the relation exhibits a high volume of missing edges.

- We propose a novel framework SmGNN which learns the relevant representation from other types of relations. And this representation is used to augment the graph structure of the sparse layer to learn expressive representation for missing link prediction.

- Experiments on real-world multi-layer graphs demonstrate the effectiveness of the proposed framework SmGNN.

## 2 Related Work

**Graph Neural Networks.** Graph Neural Networks (GNNs) are popular approaches for node representation learning on graphs. Generally, existing GNNs can be categorized into two types: spectral-based (Bruna et al., 2013; Kipf & Welling, 2016a; Tang et al., 2019; He et al., 2021; Wang & Zhang, 2022; He et al., 2022) and spatial-based (Veličković et al., 2017; Hamilton et al., 2017; Veličković et al., 2018; Gao et al., 2018; Zhang et al., 2018; Ying et al., 2018; Xiao et al., 2021). Spectral-based GNNs use graph signal processing and apply convolutional operations to graph data in the spectral domain (Bruna et al., 2013). GCN, which uses a first-order approximation, is an example of a spectral-based GNN (Kipf & Welling, 2016a). Spatial-based GNNs, on the other hand, aggregate information from neighboring nodes to update the representation of a given node. GAT (Veličković et al., 2017) is an example of a spatial-based GNN that utilizes attention mechanisms to update node representations using different weights from neighbor nodes.

**Multi-layer Graph Neural Networks.** The multi-layer graph (Li et al., 2018), also referred to as a multiplex (Cen et al., 2019; Park et al., 2020), multi-view (Qu et al., 2017) or multi dimensional graph (Ma et al., 2019), considers multiple relationships among nodes. Various approaches have been proposed to handle multi-layer graphs. For example, MVE (Qu et al., 2017) and HAN (Wang et al., 2019) utilize attention mechanisms to combine embeddings from different views. mGCN (Ma et al., 2019) models interactions within and across views for node classification. Other recent research has introduced various methods focused on learning node embeddings, which are subsequently used in node clustering and classification tasks (Fu et al., 2020; Sun et al., 2019). For instance, VANE (Fu et al., 2020) employs adversarial training to improve the comprehensiveness and robustness of node representation learning. These techniques enable better handling of the complexity of multi-layer graphs and can lead to improved performance in various tasks. Moreover, contrastive learning is also adopted to learn expressive representation for multi-layer graphs (Jing et al., 2021). For instance, HDMI (Jing et al., 2021) learns network embeddings for multi-layer networks by utilizing high-order mutual information and a fusion module based on an attention mechanism. X-GOAL (Jing et al., 2022) propagates information across different layers of multiplex heterogeneous graphs using a GOAL framework for each layer and an alignment regularization technique. However, in the real world, there may be many missing edges for some relations in multi-layer graphs due to privacy issues. Therefore, in this paper, we study a novel problem of link prediction on sparse layers in multi-layer graphs by leveraging other layers' information.

**Link Prediction.** Link prediction has wide applications such as social networks (Adamic & Adar, 2003) and knowledge graphs (Nickel et al., 2015). Existing link prediction methods can be broadly classified into two categories: heuristic-based (Lü et al., 2009; Newman, 2001; Adamic & Adar, 2003; Katz, 1953) and representation learning-based (Acar et al., 2009; Kipf & Welling, 2016b; Zhang & Chen, 2018; Pan et al., 2022). Heuristic-based approaches include methods such as the common-neighbor index (CN) (Newman, 2001), Adamic-Adar (AA) (Adamic & Adar, 2003), Katz (Katz, 1953), and rooted PageRank (PR) (Brin & Page, 1998), which are calculated based on number of common neighbors or local-neighbor similarity of the target nodes. However, these heuristic-based methods make strong assumptions and cannot be generalized to different types of graph data. Representation learning-based approaches, on the other hand, first learn the node representations and then use the dot product between two node representations to predict the likelihood of a link between them. Graph Neural Networks (GNNs) are widely used to learn node-level representations that capture both the topology structure and node feature information, achieving state-of-the-art performance in link prediction (Kipf & Welling, 2016b; Zhang & Chen, 2018; Pan et al., 2022). For example, VGAE (Kipf & Welling, 2016b) applies GNNs to learn node representations, followed by a simple

inner product decoder to predict link probabilities. SEAL (Zhang & Chen, 2018) extracts subgraphs to predict links between nodes. However, there is no research about exploring the issue of having sparse edges for the link prediction task of multi-layer graphs.

Our work is inherently different from existing works: (**i**) existing works on link prediction mainly focus on single-layer graphs; while we study a novel problem of link prediction for sparse layers on multi-layer graphs; (**ii**) we propose a novel framework that can leverage other layers to facilitate link prediction in the sparse layer.

## 3 Problem Definition and Notations

We use $\mathcal{G} = \{\mathcal{V}, \mathcal{E}_1, \ldots, \mathcal{E}_L\}$ to denote an $L$-layer attributed graph, where $\mathcal{V} = \{v_1, \ldots, v_N\}$ is the set of $N$ nodes, $\mathcal{E}_l \subseteq \mathcal{V} \times \mathcal{V}$ is the set of edges in the $l$-th layer. $\mathbf{X}$ is the node attribute matrix with $\mathbf{X}[j,:] \in \mathbb{R}^{1 \times d}$ being node attribute vector for node $v_j$. $\mathbf{A}^l$ is the adjacency matrix for the $l$-th layer. $A_{i,j}^l = 1$ if nodes $v_i$ and $v_j$ are connected in layer $l$, otherwise $A_{i,j}^l = 0$. In the real world, the collected multi-layer graph might have a very sparse layer with many unobserved edges due to various issues such as privacy issues or difficulty in obtaining the edges in that layer (Bahulkar et al., 2018); while it is important to predict the missing links in that layer. As other layers might have a correlation with the sparse layer, they can be utilized to predict the missing links in the sparse layer. Therefore, in this paper, we propose to leverage other layers' information to predict missing edges for the sparse layer. *Without loss of generality, we treat the L-th layer as the target sparse layer for link prediction.* Note that the choice of the target sparse layer can be flexible for this problem. We denote the observed edges set of the sparse layer as $\mathcal{E}_L^O$ and a set of node pairs $\mathcal{E}_L^U$. $\mathcal{E}_L^U$ includes node pairs with unobserved links and node pairs without links. The problem of link prediction for the sparse layer can be formulated as a classification problem on determining whether there is a link between $v_i$ and $v_j$ for each node pair $(v_i, v_j) \in \mathcal{E}_L^U$ given observed edges $\mathcal{E}_L^O$ and other layers' edge information $\{\mathcal{E}_1, \ldots, \mathcal{E}_{L-1}\}$, where $\mathcal{E}_l$ is the set of edges for $l$-th layer. Note that $\mathbf{A}^L$ is the adjacency matrix describing $\mathcal{E}_L^O$ for the sparse layer. In the following part, we call the target layer (index $L$) with many missing edges as the sparse layer, and other layers are denoted as dense layers. With the notations above, the problem is formally defined as:

*Given a multi-layer graph $\mathcal{G} = (\mathcal{V}, \mathcal{E}_1, \ldots, \mathcal{E}_L, \mathbf{X})$, where layer $L$ is the target sparse layer for link prediction, we denote the set of observed edges for layer $L$ as $\mathcal{E}_L^O$ and the set of unobserved or nonexistent edges as $\mathcal{E}_L^U$. Our task is to use the observed edge sets $\{\mathcal{E}_1, \ldots, \mathcal{E}_{L-1}\}$ together with $\mathcal{E}_L^O$ to learn a link predictor $g_\theta$ to accurately predict the existence of links in $\mathcal{E}_L^U$.*

## 4 Proposed Method

In this section, we introduce the proposed Sparse Layer Reconstruction Multi-layer Graph Neural Network (SmGNN) framework designed for the sparse layer's link prediction of multi-layer graphs. An illustration of the proposed framework is shown in Figure 2. As the sparse layer doesn't have enough edge information, directly training an edge predictor on the sparse layer would result in poor performance. Thus, SmGNN focuses on extracting relevant link information from other layers, which can then be utilized to predict missing links in a sparse layer. The primary objective of SmGNN is to leverage this relevant information to improve the link reconstruction performance for the sparse layer. Specifically, SmGNN introduces the concept of learning relevant representations across layers with the Relevant Information Encoder module. SmGNN utilizes supervised signals to reconstruct the structural information of both the dense layers and the sparse layer, ensuring that the relevant representations contain sufficient information from dense layers, which are relevant to the characteristics of the sparse layer. Finally, the relevant representation is utilized to enhance the structure of the sparse layer, resulting in a more expressive representation by learning from the augmented graph structure. This enriched representation is then used to effectively reconstruct the missing links within the sparse layer, improving the performance of link prediction. Next, we introduce the details.

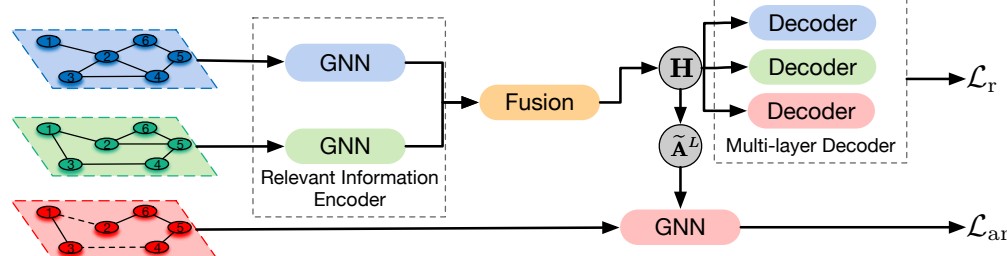

Figure 2: An overview of the proposed SmGNN. We use a multi-layer graph with $L = 3$ layers as an example. The first two layers contain abundant observed edges, while the last layer, referred to as the sparse layer, has a significant number of missing edges. SmGNN aims to extract pertinent information from the other layers and utilize it to reconstruct the hidden interactions within the sparse layer.

### 4.1 Relevant Information Encoder

As mentioned above, one main challenge is that the sparse layer does not have enough edges to learn good node representations and to train a good link predictor. Fortunately, the structural information from other layers (referred to as dense layers) can be leveraged to identify and predict missing links within the sparse layer $L$. As not all layers are that highly correlated with layer $L$, those not highly relevant information might introduce noise. Hence, it is crucial to effectively gather and utilize the *relevant information* from some dense layers to predict the links within the sparse layer. To achieve this goal, we propose a relevant information encoder to learn the relevant node representation matrix $\mathbf{H}$, which captures the informative aspects present in the dense layers, enabling the reconstruction of missing links within the sparse layer.

Specifically, as graphs contain both feature information and structural information, we propose to use $L-1$ Graph Neural Network (GNN) encoders to learn node representations from the $L-1$ layers, respectively. The reason why we don't use the shared encoder is that multi-layer graphs often exhibit heterogeneity, with each layer representing different types of connections and possessing distinct structural properties. By using separate GNN encoders, we can better capture and differentiate these various relationships. Each GNN encoder is responsible for extracting valuable information from its respective dense layer, encompassing both the feature information and the structural characteristics. For each layer $l \in [1, L-1]$, we utilize a GNN model with parameters $\theta_l$ as encoder to learn the node representation as:

$$\mathbf{H}^l = GNN_{\theta_l}(\mathbf{X}, \mathbf{A}^l), \quad l \in [1, L-1], \tag{1}$$

where $\mathbf{H}^l$ represents $l$ layer's node representation matrix.

With the representations, we will then extract useful information from these node representations, which can be used to predict missing edges in the sparse layer $L$. However, in a multi-layer graph, some layers may have a higher correlation with the sparse layer, while others may capture less important information or not be relevant to the sparse layer. Therefore, it is important to assign larger weights to relevant layers to capture important information and small weights to irrelevant layers to reduce noisy information, which can better help link prediction in the sparse layer $L$. Specifically, for each node $v_i$, we calculate a weight vector $a_i \in \mathbb{R}^{L-1}$, with $a_{i,j}$ denoting the relative importance of layer $l$'s information in contributing to the final relevant representation of $v_i$. Concretely, we calculate $\mathbf{a}_i$ as:

$$a_i = \text{Softmax}([\mathbf{H}_i^1||...||\mathbf{H}_i^{L-1}]\mathbf{W}_o + \mathbf{b}_o) \tag{2}$$

where $\mathbf{W}_o$ and $\mathbf{b}_o$ are learnable parameters, $\mathbf{H}_i^l$ is the representation vector of the node $v_i$ from the GNN encoder for the layer $l$ and $\text{Softmax}(\cdot)$ represents the Softmax function. $[\mathbf{H}_i^1||...||\mathbf{H}_i^{L-1}]$ means the concatenation operations of all vectors $\mathbf{H}_i^l$ with $l \in [1, L-1]$. The relevant representation of node $v_i$ is obtained by taking a weighted sum of the hidden states from all layers in the graph as:

$$\mathbf{H}_i = a_{i,1} \cdot \mathbf{H}_i^1 + ... + a_{i,L-1} \cdot \mathbf{H}_i^{L-1}, \tag{3}$$

where $\mathbf{H}_i$ is the relevant representation for node $i$ and the representation matrix can be denoted as $\mathbf{H}$.

## 4.2 Multi-layer Decoder

The relevant representation should capture the essential structural characteristics and patterns present in the dense layers, and can also be used to accurately explore the missing links within the sparse layer. To achieve this goal, we propose supervision signals for learning representation $\mathbf{H}$. Specifically, we guide the learning process of the relevant representation $\mathbf{H}$ by reconstructing the edges across both dense layers and the sparse layer. This approach enables the relevant representation to capture valuable structural information present in the dense layers, which in turn can be utilized to reconstruct existing links within the sparse layer. To reconstruct $\mathbf{A}^l$, we first adopt an MLP parametrized by $\theta'_l$ to project $\mathbf{H}$ to the feature space for $l$-th layer as:

$$\mathbf{F}^l = \mathrm{MLP}_{\theta'_l}(\mathbf{H}), l \in [1, L] \tag{4}$$

Then the $l$-th layers adjacency matrix is reconstructed as:

$$\hat{\mathbf{A}}^l = \mathrm{Sigmoid}(\mathbf{F}^l(\mathbf{F}^l)^T) \tag{5}$$

where $\hat{\mathbf{A}}^l$ is the reconstructed adjacency matrix for layer $l$ with $\hat{A}^l_{ij}$ denoting the link probability between node $v_i$ and $v_j$ in layer $l$.

We want the reconstructed $\hat{\mathbf{A}}^l$ to be close to $\mathbf{A}^l$. However, since the majority of node pairs are unconnected, most of the elements in adjacency matrix $\mathbf{A}^l$ are 0. Directly using cross-entropy loss between $\hat{\mathbf{A}}^l$ and $\mathbf{A}^l$ would make the loss dominated by missing link. To avoid the missing links dominating the loss function, following Zhang & Chen (2018), we adopt negative sampling to alleviate this issue. We treat each linked pair in $\mathcal{E}^O_l$ as positive samples for the layer $l$. For each positive sample, we randomly sample one unlinked pair as the negative sample. The set of randomly selected unlinked pairs is denoted as $\mathcal{E}^N_l$ for the layer $l$. Then, we treat link prediction as a binary classification problem to predict positive and negative samples. To guide the relevant representation to learn useful information for reconstructing the sparse layer, we can minimize the edge prediction error for all layers based on the learned representation as:

$$\mathcal{L}_r = \sum_{l=1}^{L} \Big( \sum_{e_{ij} \in \mathcal{E}^O_l} -\log \hat{A}^l_{i,j} + \sum_{e_{ij} \in \mathcal{E}^N_l} -\log \Big( 1 - \hat{A}^l_{i,j} \Big) \Big). \tag{6}$$

By minimizing $\mathcal{L}_r$, the encoded representation $\mathbf{H}$ can learn relevant information in other layers and this information can also be used to reconstruct the structural information of that sparse layer $L$.

## 4.3 Sparse Layer Augmentation

Though we learn the relevant representation from dense layers to explore missing links in the sparse layer, there is another unsolved challenge, i.e., how to use this relevant representation to enhance the link prediction process of the sparse layer. In this subsection, we will illustrate how SmGNN learns the relevant representation $\mathbf{H}$ to enhance the structural information of the sparse layer, thereby enabling the learning of more expressive representations for edge prediction. Specifically, to reconstruct edge information in the sparse layer $L$, current Graph Autoencoder (GAE) (Kipf & Welling, 2016b) uses GNNs to encode the edges and feature information into representation vectors. Subsequently, an edge prediction task is utilized to learn these representation vectors that can predict missing links within the graph. If we use existing link information in the sparse layer, we might not learn expressive representation which can accurately predict the missing edges. This is because the majority of node pairs in the sparse layer are not connected, where GNNs can not learn informative representation and lack supervision signals.

To resolve this problem, we propose to augment the original graph structure of layer $L$ via $\mathbf{H}$ because $\mathbf{H}$ contains relevant information from dense layers, which can be used to reconstruct the sparse layer. However, it is time-consuming to augment the original graph by considering all possible node pairs without links for each training epoch. Before the training process, for each node $v_i$, we find $K$ nodes $v_j \in \mathcal{V}$ that are likely to have links in layer $L$ to construct candidate node pairs, with the constraint that $(v_i, v_j) \notin \mathcal{E}^O_L$ and $j \neq i$. By the assumption that node pairs having similar features are more likely to be connected in the sparse layer, we select top $K$ nodes of $v_i$ based on the similarity between the original feature vector $\mathbf{X}_i$ of node $v_i$ and the

original feature vector $\mathbf{X}_j$ of nodes $v_j \in \mathcal{V}$, where $i \neq j$. Though the similarity of the original features alone may not accurately predict missing edges in the sparse layer, it can still be useful in identifying potentially connected node pairs. This approach can help save time complexity by narrowing down the search space for edge augmentation for the sparse layer. We do the same operation for all nodes. These selected node pairs construct a new set $\tilde{\mathcal{E}}$ for the sparse layer $L$, where the size of it is $|\tilde{\mathcal{E}}| = K \cdot N$. Note that we will make node pairs in this set have no repetition by deleting repeated pairs in the set.

Then, for each epoch during training, we select node pairs from the candidate set $\tilde{\mathcal{E}}$ to build edges based on the similarity of the relevant representation $\mathbf{H}$, which are used to augment the sparse layer. As the original feature $\mathbf{X}$ can also provide valuable information, for each node pair $(v_i, v_j) \in \tilde{\mathcal{E}}$, we propose to combine both relevant representation $\mathbf{H}$ from other layers and original feature $\mathbf{X}$ to calculate the similarity between node pairs $v_i$ and $v_j$ to augment graph structure of the sparse layer as:

$$S_{ij} = \cos([\mathbf{X}_i||\mathbf{H}_i], [\mathbf{X}_j||\mathbf{H}_j]), \tag{7}$$

where $\cos(\cdot)$ denotes the cosine similarity function that calculates the pair-wise similarity. For $(v_i, v_j) \notin \tilde{\mathcal{E}}$, we simply set $S_{ij} = 0$. We need to use $S_{ij}$ to represent the probability of the existence of links. The element in an adjacency matrix (computed from a metric) is supposed to be non-negative but $s_{ij}$ ranges between $[-1, 1]$ which could produce negative value in the adjacency matrix. Also, preserving all similarity values in $\mathbf{S}$ might introduce noise (i.e., unimportant edges). To address this, we create a sparse similarity matrix $\mathbf{S}^{sp}$ by setting elements smaller than a non-negative threshold $\epsilon$. In our experiments, we found that the specific value of $\epsilon$ doesn't significantly affect the results. Hence, We set $\epsilon$ to 0, ensuring non-negativity in the augmented adjacency matrix $\widetilde{\mathbf{A}}^L$. Then, we obtain the augmented graph as:

$$\widetilde{\mathbf{A}}^L = \alpha \cdot \mathbf{S}^{sp} + \mathbf{A}^L \tag{8}$$

where $\alpha$ is the hyperparameter to control the weight for the augmented graph. With the augmented adjacency matrix, we first use the GNN encoder $GNN_{\theta_L}$ to learn node representation as:

$$\mathbf{H}^L = [GNN_{\theta_L}(\mathbf{X}, \widetilde{\mathbf{A}}^L)||\mathbf{H}], \tag{9}$$

where $\theta_L$ is the learnable parameter for the sparse layer $L$. In the above equation, the augmented matrix $\widetilde{\mathbf{A}}^L$ has the knowledge from sparse similarity matrix $\mathbf{S}^{sp}$ which contains information from other layers. It could help learn better representations compared to Eq. 1. we also incorporate the relevant representation from dense layers with the learned representation from the sparse layer, thereby further enhancing the expressive power of the resulting representation $\mathbf{H}^L$ for the sparse layer. With $\mathbf{H}^L$, we can complete the adjacency matrix of the sparse layer $L$ as:

$$\bar{\mathbf{A}}^L = \text{Sigmoid}(\mathbf{H}^L(\mathbf{H}^L)^T), \tag{10}$$

## 4.4 Adaptive Sparse Layer Reconstruction

With the predicted adjacency matrix $\bar{\mathbf{A}}^L$, one straightforward way is to employ the edge prediction loss on observed edges of $\mathbf{A}^L$ to guide the learning process of these representations as introduced in Eq. (6). This task helps refine the node representations to predict missing links. However, the graph of layer $L$ has a limited number of observed edges. Consequently, relying solely on the original observed graph structure as supervised signals for edge prediction makes it challenging for the learned node representations to accurately predict missing links in the sparse layer. To provide more supervised signals, we select top $R$ edges based on weight scores in $\mathbf{S}$ for every $T$ epoch. Then, we add these edges into the observed edges set $\mathcal{E}_L^O$ to obtain a new set $\widetilde{\mathcal{E}}_L^O$. Moreover, as discussed in Section 4.2, we also sample an equal number of negative samples to form a set $\widetilde{\mathcal{E}}_L^N$. Then the loss for edge prediction of the layer $L$'s with the augmented edge set is:

$$\mathcal{L}_{ar} = \sum_{e_{ij} \in \widetilde{\mathcal{E}}_L^O} -\log \bar{A}_{i,j}^L + \sum_{e_{ij} \in \widetilde{\mathcal{E}}_L^N} -\log\left(1 - \bar{A}_{i,j}^L\right), \tag{11}$$

where $\bar{A}_{i,j}^L$ is a predicted probability for the edge between $v_i$ and $v_j$ in Eq. (10) in the layer $L$. The final objective of SmGNN is:

$$\min_{\Theta} \mathcal{L} = \mathcal{L}_{ar} + \gamma \mathcal{L}_r, \tag{12}$$

Table 1: Link Prediction Performance AUC and Average Precision (AP). The best and second-best performances under each layer of the dataset are marked with boldface and underlined, respectively. $l'$ represents the index of the sparse layer.

| Metric | Method | Epinions | | | | | Amazon | | | IMDB | |
|---|---|---|---|---|---|---|---|---|---|---|---|
| | | $l'$=1 | $l'$=2 | $l'$=3 | $l'$=4 | $l'$=5 | $l'$=1 | $l'$=2 | $l'$=3 | $l'$=1 | $l'$=2 |
| AUC | MLP | 80.22±0.02 | 78.95±0.06 | 84.33±0.03 | 79.21±0.04 | 80.32±0.21 | 81.23±0.13 | 82.44±0.09 | 83.31±0.07 | 70.64±0.12 | 71.99±0.20 |
| | GCN | 99.43±0.01 | 95.83±0.17 | 99.55±0.02 | 91.69±0.23 | 87.00±0.38 | 96.26±0.22 | 98.65±0.29 | 88.08±0.25 | 85.88±0.16 | 81.86±0.15 |
| | GCN-C | 99.58±0.03 | 96.07±0.09 | 99.61±0.04 | 92.41±0.35 | 93.39±0.11 | 94.90±0.18 | 98.38±0.04 | 89.04±0.27 | 81.96±0.22 | 84.12±0.16 |
| | mGCN | 99.50±0.08 | 95.98±0.03 | 99.70±0.03 | 92.83±0.81 | 90.04±0.96 | 95.70±0.12 | 97.48±0.03 | 90.10±0.16 | 86.78±0.14 | 83.21±0.27 |
| | HDMI | 96.32±0.02 | 94.43±0.02 | 98.33±0.06 | 90.21±0.11 | 88.32±0.33 | 94.57±0.24 | 96.12±0.28 | 85.31±0.30 | 83.41±0.22 | 76.55±0.10 |
| | WP | 99.55±0.03 | 96.11±0.03 | 99.67±0.05 | 93.07±0.13 | 91.86±0.12 | 96.58±0.04 | 96.58±0.04 | 89.30±0.13 | 86.70±0.42 | **87.30±0.32** |
| | X-GOAL | 97.21±0.05 | 95.71±0.07 | 98.51±0.03 | 92.33±0.08 | 90.45±0.04 | 95.00±0.13 | 98.21±0.18 | 87.25±0.23 | 84.91±0.13 | 78.44±0.39 |
| | SmGNN | **99.64±0.03** | **96.20±0.06** | **99.73±0.01** | **93.44±0.10** | **95.75±0.36** | **96.75±0.21** | **98.75±0.07** | **94.33±0.16** | **86.81±0.09** | 83.28±0.12 |
| AP | MLP | 90.32±0.05 | 81.65±0.08 | 87.67±0.03 | 80.33±0.02 | 81.92±0.36 | 80.57±0.04 | 81.73±0.05 | 81.31±0.06 | 72.33±0.08 | 70.86±0.22 |
| | GCN | 99.57±0.05 | 96.21±0.07 | 99.70±0.04 | 93.19±0.11 | 86.71±0.21 | 96.72±0.10 | 97.86±0.09 | 91.23±0.23 | **90.97±0.17** | 85.82±0.10 |
| | GCN-C | 99.59±0.04 | 97.29±0.06 | 99.83±0.05 | 96.43±0.20 | 95.32±0.15 | 94.03±0.22 | 98.03±0.02 | 84.05±0.28 | 81.59±0.41 | 85.86±0.01 |
| | mGCN | 99.21±0.14 | 95.81±0.07 | 99.36±0.02 | 94.42±0.32 | 93.03±0.16 | 96.20±0.23 | 97.59±0.03 | 89.43±0.31 | 89.70±0.25 | 85.97±0.24 |
| | HDMI | 96.73±0.16 | 94.22±0.08 | 97.38±0.04 | 93.05±0.17 | 92.19±0.20 | 94.11±0.31 | 95.33±0.14 | 85.32±0.42 | 80.53±0.24 | 76.44±0.13 |
| | WP | 99.72±0.90 | 97.33±0.06 | 99.79±0.02 | 95.66±0.22 | 93.94±0.26 | 96.20±0.16 | 98.11±0.02 | 86.03±0.10 | 90.02±0.41 | 85.40±0.41 |
| | X-GOAL | 97.04±0.06 | 95.23±0.04 | 97.92±0.05 | 94.53±0.11 | 92.42±0.16 | 94.77±0.32 | 95.54±0.04 | 87.22±0.26 | 82.99±0.35 | **86.72±0.40** |
| | SmGNN | **99.77±0.03** | **98.14±0.05** | **99.85±0.02** | **97.35±0.12** | **95.75±0.19** | **96.74±0.15** | **98.80±0.02** | **93.91±0.13** | 88.40±0.14 | 86.63±0.06 |

Table 2: Link prediction Performance on Hits@500 and Hits@1000.

| Metric | Method | Epinions | | | | | Amazon | | | IMDB | |
|---|---|---|---|---|---|---|---|---|---|---|---|
| | | $l'$=1 | $l'$=2 | $l'$=3 | $l'$=4 | $l'$=5 | $l'$=1 | $l'$=2 | $l'$=3 | $l'$=1 | $l'$=2 |
| Hits@500 | MLP | 21.73±0.04 | 15.32±0.06 | 27.98±0.09 | 29.88±0.02 | 25.64±0.05 | 8.93±0.03 | 9.17±0.04 | 8.64±0.07 | 33.98±0.13 | 33.02±0.20 |
| | GCN | 94.76±0.17 | 50.01±0.23 | 92.42±0.18 | 62.00±0.17 | 50.21±0.22 | 53.96±0.19 | 50.14±0.03 | 72.01±0.07 | 70.63±0.21 | 45.32±0.11 |
| | GCN-C | 92.84±0.29 | 50.51±0.26 | 92.36±0.13 | 64.20±0.15 | 52.44±0.10 | 40.41±0.09 | 38.03±0.11 | 58.50±0.09 | 58.02±0.10 | 41.81±0.03 |
| | mGCN | 72.63±0.23 | 45.68±0.28 | 73.03±0.30 | 53.39±0.17 | 43.76±0.14 | 45.95±0.07 | 26.99±0.08 | 57.30±0.13 | 36.05±0.23 | 23.15±0.27 |
| | HDMI | 65.42±0.22 | 34.21±0.24 | 70.03±0.18 | 42.55±0.14 | 41.31±0.09 | 39.22±0.16 | 30.23±0.18 | 37.12±0.09 | 33.66±0.17 | 40.02±0.09 |
| | WP | 93.22±0.14 | 50.22±0.29 | 92.27±0.16 | 64.42±0.05 | 58.34±0.06 | 56.06±0.26 | 50.22±0.08 | 77.08±0.13 | 70.01±0.45 | 49.01±0.45 |
| | X-GOAL | 71.22±0.08 | 41.44±0.13 | 73.62±0.11 | 43.23±0.10 | 43.32±0.12 | 40.50±0.08 | 27.75±0.33 | 50.40±0.22 | 35.47±0.32 | 42.35±0.45 |
| | SmGNN | **95.27±0.18** | **51.02±0.16** | **93.36±0.26** | **66.40±0.08** | **60.87±0.11** | **60.76±0.04** | **50.62±0.09** | **82.19±0.10** | **71.38±0.08** | **49.16±0.13** |
| Hits@1000 | MLP | 28.65±0.03 | 26.33±0.05 | 40.21±0.07 | 55.34±0.02 | 32.99±0.08 | 16.04±0.03 | 16.15±0.07 | 15.37±0.06 | 35.87±0.14 | 49.28±0.13 |
| | GCN | 97.39±0.21 | 62.61±0.20 | 95.44±0.31 | 71.53±0.26 | 64.29±0.20 | 72.32±0.16 | 62.62±0.19 | 81.25±0.08 | 81.50±0.20 | 52.45±0.26 |
| | GCN-C | 97.00±0.28 | 60.08±0.19 | 96.06±0.38 | 73.11±0.26 | 56.36±0.23 | 57.87±0.14 | 50.19±0.16 | 84.97±0.03 | 73.11±0.36 | 53.29±0.39 |
| | mGCN | 76.32±0.16 | 53.92±0.14 | 77.59±0.22 | 60.98±0.26 | 59.10±0.18 | 55.30±0.09 | 37.19±0.06 | 67.19±0.02 | 39.94±0.17 | 27.72±0.13 |
| | HDMI | 70.82±0.16 | 39.88±0.24 | 76.49±0.42 | 54.76±0.19 | 49.85±0.20 | 43.34±0.11 | 27.88±0.19 | 54.11±0.09 | 36.90±0.26 | 37.82±0.06 |
| | WP | 97.62±0.18 | 62.24±0.11 | 96.21±0.13 | 74.95±0.0.13 | 66.98±0.16 | 67.68±0.22 | 61.09±0.23 | 86.54±0.13 | **83.31±0.44** | 52.92±0.26 |
| | X-GOAL | 74.32±0.06 | 40.54±0.03 | 78.45±0.19 | 56.63±0.08 | 54.43±0.12 | 53.26±0.70 | 43.26±0.24 | 60.22±0.11 | 39.77±0.33 | 43.31±0.22 |
| | SmGNN | **97.80±0.14** | **62.48±0.20** | **96.77±0.32** | **75.39±0.25** | **70.70±0.27** | **72.40±0.02** | **63.63±0.14** | **93.20±0.07** | 82.68±0.16 | **56.51±0.13** |

where $\Theta$ is the set of learnable parameters $\{\theta_1, ..., \theta_L, \theta'_1, ..., \theta'_L, \mathbf{W}_o, \mathbf{b}_o\}$ and $\gamma$ represents a weight parameter that controls the loss contribution of the supervised information for the learning process to acquire a relevant representation. We put the training algorithm in Appendix A and time complexity analysis in Appendix D.

## 5 Experiments

In this section, we conduct experiments on real-world multi-layer graphs to demonstrate the effectiveness of SmGNN. In particular, we aim to answer the following research questions: (**RQ1**) Can SmGNN provide accurate link predictions for sparse layers? (**RQ2**) Can SmGNN deal with different levels of the edge sparsity issue? (**RQ3**) What are the contributions of each component for SmGNN?

### 5.1 Datasets

We conduct experiments on three publicly available real-world multi-layer:

- **Epinions** (Ma et al., 2019): Epinions is an online product review site, where users can post reviews of various products and rate the usefulness of reviews posted by other users. Also, users on this platform can form trust and distrust relations. It forms a five-layer graph based on 5 different relationships: 1) co-review: two users review common products; 2) helpfulness-rating: a user rates the reviews written by the other user; 3) co-rating: two users rate some common reviews; 4) trust relation between users; and 5) distrust relation between users.

- **Amazon** (He & McAuley, 2016): It is sourced from Amazon.com and represents a network of items, with each node representing an item. It forms a three-layer graph based on three different relations: (1) also-view: two items are viewed by the same customer; (2) also-bought: two items are bought by the same customer (3) bought-together: two items are bought by a customer at the same time.

- **IMDB**: The IMDB [1] dataset contains 3,550 movies with two types of relations, including movie-actor-movie and movie-director-movie. The attribute of each movie is a 1,007-dimensional bag-of-words representation of its plots.

The statistics of these datasets are summarized in Table 4 in Appendix B. For each experiment, one layer is chosen as the target layer for link prediction and the other layers are used as dense layers. For Epinions, IMDB, and Amazon, we randomly split the target layer's edges into 40%/20%/40% as train/val/test. We randomly select node pairs not in the training set as negative samples following Kipf & Welling (2016b). The number of negative samples is equal to the number of positive samples. Positive and negative samples are combined as our training, validation, and testing sets (Kipf & Welling, 2016b). The random split is conducted 5 times and average performance is reported.

## 5.2 Experimental Setup

**Baselines.** We compare SmGNN with representative and state-of-the-art methods for link prediction:

- **MLP**: It utilizes a multi-layer perceptron (MLP) with node attributes as input to predict links. It is trained by minimizing binary cross-entropy loss between the predicted link probability and the labels that denote link existence.

- **GCN** (Kipf & Welling, 2016a): GCN is one of the most popular spectral GNN models based on graph Laplacian, which has shown great performance for node classification. To adopt it for link prediction, we treat it as the encoder in the Graph Autoencoder manner only on the target sparse layer.

- **GCN-C**: As GCN can only deal with single-layer graphs, for GCN-C, we combine edge information from different layers to construct one single adjacency matrix. Then, we adopt GCN on the combined adjacency matrix for link prediction.

- **mGCN** (Ma et al., 2019): mGCN utilizes GCN to extract node embedding for each layer of the multi-layer graphs and then combine them via the attention mechanism. It shows great effectiveness in modeling relations in different layers.

- **HDMI** (Jing et al., 2021): HDMI is a baseline designed for multi-layer graphs, which adopts a new contrastive learning loss by capturing high-order information across different layers.

- **X-GOAL** (Jing et al., 2022): It includes a GOAL framework, which learns node embeddings for each graph layer, and an alignment regularization technique to jointly model and propagate information across different layers. After obtaining node embeddings for both HDMI and X-GOAL, we employ MLPs to perform binary classification, determining whether there exists an edge between pairs of nodes.

- **WP** (Pan et al., 2022): WalkPooling (WP) jointly encodes node representations and graph topology into learned topological features. Then, these features are used to enhance the representation of extracted subgraphs that are relevant to links of node pairs. It is a state-of-the-art model designed for link prediction.

**Configurations.** All experiments are conducted on a 64-bit machine with Nvidia GPU (NVIDIA RTX A6000, 1410MHz, 48 GB memory). For a fair comparison, we utilize a two-layer GCN for all methods, where the hidden dimension is set as 128. The learning rate is initialized to 0.001. Besides, all models are trained until converging, with the maximum training epoch as 1000. The implementations of all baselines are based on Pytorch Geometric or their original code. The hyperparameters of all methods are tuned on the validation set. For the hyperparameters of our model, we vary $\alpha$ as $\{0, 0.1, 0.3, 0.5, 0.7, 1\}$. $\gamma$ is varied as $\{0, 0.1, 0.3, 0.5, 0.7, 1\}$. We vary $R$ as $\{0, 100, 500, 1000, 2000\}$. $K$ is fixed as 50 for all datasets.

---

[1] https://www.imdb.com/

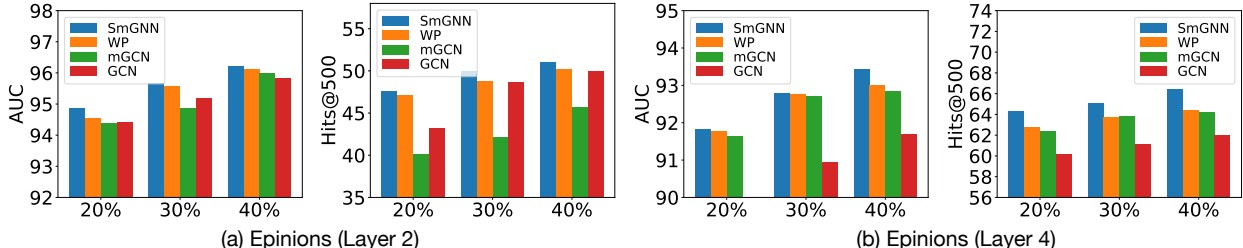

Figure 3: Experiments with different sparsity edge levels for link prediction.

**Evaluation Metrics.** Following existing works on link prediction (Kipf & Welling, 2016a; Zhang & Chen, 2018; Hu et al., 2020), we adopt AUC value, average precision, Hits@500 and Hits@1000 as the evaluation metrics. Specifically, for the evaluation metric Hits@$Q$, for each positive edge in the test set, we rank it against $|\mathcal{E}_L^O|$ randomly-sampled negative edges and count the ratio of positive edges that are ranked at $Q$-th place or above (Hits@$Q$), where $Q$ is set to 500 and 1000 and it can provide a good threshold to rate the models' performance (Hu et al., 2020).

## 5.3 Link Prediction Performance

In this subsection, we compare the performance of the proposed method with baselines for link prediction on the multi-layer graphs, which aims to answer **RQ1**. For all datasets, each experiment is conducted 5 times, and average results and standard deviations are reported in Table 1 and Table 2, where $l'$ denotes the index of the target sparse layer, while the other layers are used as dense layers. For example, $l' = 2$ for Epinions means we use layer 2 of Epinions as the sparse layer. From the table, we observe: ($i$) Compared with GCN, GCN-C can consistently improve the performance of GCN on Amazon and Epinions. It demonstrates that combining other layers' information can help to reconstruct edges for the sparse layer. However, GCN-C's performance on IMDB drops a lot compared with GCN on layer one. It means that simply combining edges from different layers may not effectively explore latent information in each layer and improve performance for link prediction. Our proposed method can further outperform both GCN-C and GCN on Amazon and Epinions, which verifies the effectiveness of our method in extracting information from other layers to reconstruct the sparse layer. ($ii$) Both X-GOAL and HDMI are state-of-the-art baselines on multi-layer graphs, which adopt contrastive learning on multi-layer graphs. With sparse edges on one layer, contrastive learning-based models may not efficiently explore latent node interactions for that sparse layer. Our model can greatly outperform them on all metrics. This is because our method can effectively capture other layers' structure information, which can be helpful in discovering missing edges in the sparse layer. ($iii$) Our model can also consistently outperform WP on Epinions and Amazon for all metrics, which is the state-of-the-art baseline for link prediction based on the structure information of one single layer. This verifies our motivation that aggregating other layers' information can help reconstruct missing edges. Also, our model can utilize other layers' information to explore latent edge information for the sparse layer.

## 5.4 Link Prediction with Various Graph Sparsity

In this subsection, we explore the effectiveness of SmGNN under various edge sparsity levels for link prediction, which answers **RQ2**. Specifically, for the target layer for link prediction, we randomly mask $x\%$ observed edges for training, 20% masked edges for validation, and the remaining edges for testing, where $x \leq 40$. We vary $x$ as $\{20, 30, 40\}$ to understand the effectiveness of SmGNN under various edge sparsity levels. For each setting, the experiment is conducted 5 times and the average performance will be reported. mGCN also proposes to aggregate other layers' information, which can also mitigate the edge sparsity issue in some layers. We also include the state-of-the-art method WP for comparison. Therefore, to assess the capability of our model in learning latent edge interactions from other layers and predicting edges for the sparse layer at various levels of edge sparsity, we present the results of SmGNN, mGCN, and the base model GCN together. As we have similar observations for other datasets, we only report the results on Epinions. The corresponding results are shown in the Figure 3. We can observe that our model can consistently outperform mGCN, GCN, and WP across various levels of edge sparsity. This consistently superior perfor-

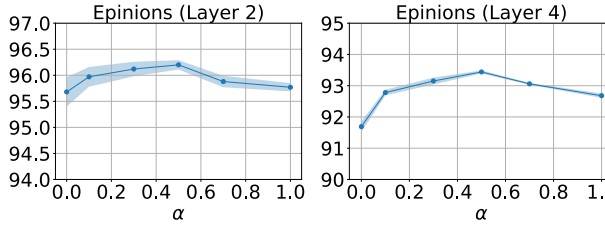

Figure 4: Hyperparameter Analysis of $\alpha$.

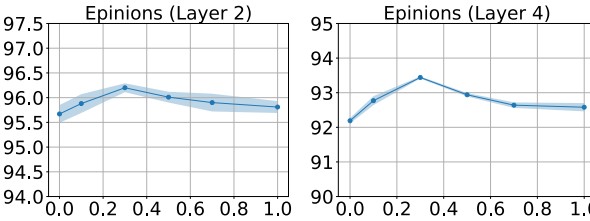

Figure 5: Hyperparameter Analysis of $\gamma$.

mance showcases the ability of SmGNN to effectively utilize the structural information from other layers to reconstruct hidden links within the sparse layer.

## 5.5 Ablation Study

To answer **RQ3**, in this section, we conduct an ablation study to evaluate the contribution of each component in SmGNN. Specifically, we consider the following ablations: (**i**) w/o augment (adj), which is a variant by removing the main component that augments the original adjacency matrix with learned graph structure from other types of relations, i.e., we replace $\widetilde{\mathbf{A}}^L$ with the original graph structure $\mathbf{A}^L$. (**ii**) w/o augment (set), which denotes the variant by removing the key com-

Table 3: Ablation Studies of SmGNN on Epinions.

| Dataset | $l'$ | Epinions | | |
|---|---|---|---|---|
| | | **2** | **4** | **5** |
| AUC | w/o augment (set) | 95.67±0.05 | 93.43±0.10 | 94.94±0.40 |
| | w/o augment (adj) | 95.93±0.05 | 93.10±0.09 | 94.69±0.23 |
| | w/o both | 95.81±0.17 | 91.69±0.23 | 87.00±0.38 |
| | w/o weight | 96.02±0.19 | 91.85±0.46 | 95.14±0.31 |
| | SmGNN | **96.20±0.06** | **93.44±0.08** | **95.75±0.36** |
| Hits@500 | w/o augment (set) | 50.40±0.22 | 63.11±0.25 | 54.31±0.18 |
| | w/o augment (adj) | 50.21±0.14 | 64.21±0.16 | 55.67±0.26 |
| | w/o both | 50.01±0.23 | 62.00±0.17 | 50.21±0.22 |
| | w/o weight | 50.90±0.15 | 63.46±0.11 | 54.15±0.32 |
| | SmGNN | **51.02±0.16** | **66.40±0.08** | **60.87±0.11** |

ponent that augments the reconstructed edge set for Eq. (11), i.e., we replace the augmented edge set $\widetilde{\mathcal{E}}_L^N$ and $\widetilde{\mathcal{E}}_L^O$ with the original observed set of the sparse layer $L$, $\mathcal{E}_L^O$ and its negative samples set $\mathcal{E}_L^N$ in Eq. (6). (**iii**) w/o both, which means that we remove these two components. (**iv**) w/o weight, which means that we immediately sum the representation from different layers without the attention mechanism in Eq. (3). The results are shown in Table 3. All experiments are conducted five times and the average performance and standard deviations are reported. We can observe that only augmenting the graph structure (w/o augment (set)) with relevant representation from other layers can help learn more expressive representation for link prediction on the sparse layer. Also, only adaptively augmenting the edge set (w/o augment (adj)) can provide more supervised signals to learn the GNN model of the sparse layer. It can be used to improve the performance of link prediction on the sparse layer. Furthermore, without the selective fusion of other types of relations by assigning different weights (w/o weight), our model fails to achieve the best performance. This finding validates our motivation to selectively extract information from other layers, improving link prediction performance for the sparse layer. Finally, the full model (last row) obtains the best performance, which illustrates that various components of SmGNN are complementary to each other.

## 5.6 Hyperparameter Analysis

The proposed method has three important hyperparameters, including $\alpha$, $\gamma$ and $R$. The analysis of $R$ is put into the Appendix of the supplementary materials. All following experiments are conducted five times and average performance are reported.

**Analysis of $\alpha$.** We conduct hyperparameter analysis of $\alpha$, which controls the weight for the learned augmented matrix of the Equation $\widetilde{\mathbf{A}}^L = \alpha \cdot \mathbf{S}^{sp} + \mathbf{A}^L$. Then, we will use the augmented $\widetilde{\mathbf{A}}^L$ to learn more expressive node representation for the sparse layer. We vary $\alpha$ as $\{0, 0.1, 0.3, 0.5, 0.7, 1\}$. The corresponding results are shown in Figure 4. We can observe that too small values of $\alpha$ (e.g., 0, 0.1) will degenerate the performance. When the values are small, it indicates that we heavily rely on the original graph structure of the sparse layer to learn the representation. Using small values can lead to poor performance because of the limited edge information available. Consequently, it becomes challenging to learn effective representations for edge prediction in the sparse layer. Moreover, large values of $\alpha$ (e.g., 0.7, 1) will also result in bad

performance. Larger values in the learned adjacency matrix $\mathbf{S}^{sp}$ imply that it covers the original structure information of the sparse layer to a greater extent. This can result in the learned representation losing its inherent structural characteristics, leading to poor performance. In general, a suitable value for $\alpha$ is in the range (0.3, 0.7).

**Analysis of $\gamma$.** We also conduct hyperparameter analysis of $\gamma$ in Eq. (12), which controls the weight for the loss of learning relevant representation from dense layers. The corresponding results are shown in the Figure 5. Large and small values of $\gamma$ will lead to bad performance. This is because large values of $\gamma$ will make the model focus too much on learning relevant representation from other layers while ignoring the loss of learning representation for missing links of the sparse layer. Moreover, when the value of $\gamma$ is small, the model tends to learn a poor relevant representation, which introduces noise in the augmented graph structure and edge set for the sparse layer. As a result, the learned representation from this augmented information leads to bad performance in link prediction for the sparse layer. Based on our experimental findings, we have observed that the optimal performance of our model is achieved within a suitable range of $\gamma$ values, typically between 0.1 and 0.5.

Another hyperparameter $K$ is only used for reducing the time consumption of our model and is not relevant to our main contributions. Hence, we fix $K$ as 50 for all datasets of our experiments.

### 5.7 Visualization

We conduct the visualization experiment for SmGNN. We aim to verify the effectiveness of the weights in Eq. (2) in capturing correlations between the sparse layers and other layers. The correlations between nodes can be measured by evaluating their similarity in local graph structures. Nodes with more similar local graph structures are likely to exhibit stronger correlations (Wang et al., 2016). To reflect the correlations between the structure of nodes in the sparse layer

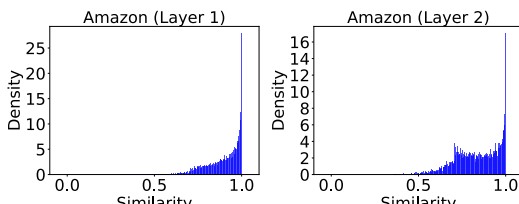

Figure 6: Visualization for SmGNN.

and the structure of nodes in other layers, we calculate the similarity based on their respective local structures. As the training part of the target layer is sparse (40%), which might not truly reflect the correlation, we use the raw graph to calculate correlations. Specifically, we denote the $m$-hop neighbors set of the node $v_i$ in the layer $l$ as $\mathcal{N}_i^l$. Then, a vector $\mathbf{p}_i^l \in \mathbb{R}^n$ is used to denote the local structure information of $v_i$, where $p_{ij}^l = 1$ if $j \in \mathcal{N}_i^l$, otherwise $p_{ij}^l = 0$. To calculate the correlation between the sparse layer $l'$ and other layers $b \in \mathcal{U}$ ($\mathcal{U} = \{1, ..., L\} / \ l'$) for the node $v_i$, we calculate the cosine similarity between $p_{ij}^{l'}$ and $p_{ij}^b$, which is denoted as $s_{i,b} = \cos(\mathbf{p}_i^{l'}, \mathbf{p}_i^b)$ and a vector $\mathbf{s}_i \in \mathbb{R}^{L-1}$ represents structure similarity between the sparse layer and other layers for the node $v_i$. To verify whether the learned weights capture local structure correlations across layers, we calculate the cosine similarity between this structure similarity vector $\mathbf{s}_i$ and the learned weight vector $\mathbf{a}_i$ for each node $v_i \in \mathcal{V}$. Finally, we visualize the distribution of these cosine similarity values with $m = 1$ and the corresponding results are in Figure 6. We can observe that the majority of cosine similarity values exceed 0.5, with a significant number of values approaching 1. This high similarity between the learned $\mathbf{a}_i$ and the structure similarity $\mathbf{s}_i$ indicates a strong correlation. This observation indicates that the learned weights for fusing information from other layers in our model effectively capture the structural correlations across layers. Consequently, the relevant information from these layers can be used to improve link prediction in the sparse layer.

## 6  Conclusion

In this paper, we propose a novel framework SmGNN for predicting a large number of missing links in sparse layers of multi-layer graphs. By selectively fusing relevant information from other layers, our model learns representations that capture the specific characteristics of the sparse layer while incorporating valuable insights from other layers. Additionally, we enhance the graph structure of the sparse layer by leveraging node similarity information based on the relevant representation. Through extensive experiments on real-world datasets, we validate the effectiveness of SmGNN in improving link prediction performance.

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

Table 4: Statistics of Datasets. $l$ means the index of the layer for multi-layer graphs.

| Dataset | # Edges | | | | | # Nodes | # Features | # Layers |
|---|---|---|---|---|---|---|---|---|
| | $l = 1$ | $l = 2$ | $l = 3$ | $l = 4$ | $l = 5$ | | | |
| Epinions | 239,636 | 338,628 | 301,886 | 79,029 | 11,129 | 338,628 | 128 | 5 |
| Amazon | 266,237 | 1,104,257 | 16,305 | - | - | 7,621 | 1,508 | 3 |
| IMDB | 66,428 | 13,788 | - | - | - | 3,550 | 1,007 | 2 |

## A  Training Algorithm

The training algorithm of our framework is given in Algorithm 1. We first extract relevant representation from other layers via Eq. (3). Then, we use supervised signals of reconstructing the sparse layer and other layers' structure information, which can guide the relevant representation to learn latent link information from other layers. Furthermore, we utilize the Graph Autoencoder framework to learn representations aimed at predicting missing links in the sparse layer. However, due to the limited edge information available in the sparse layer, the learning process is impeded in learning expressive representations for the missing links. Therefore, in line 5, we employ the relevant representation to enhance the adjacency matrix of the sparse layer, enabling the learning of more expressive representations. Additionally, in line 6, we further enhance the observed edges set to provide additional supervised signals for link prediction, refining the learned representation for the sparse layer. Then, we optimize the loss function to learn relevant representations of other layers and the representation of the sparse layer for link prediction of the sparse layer. Finally, we obtain a model which can predict missing edges for a specific layer in multi-layer graphs that exhibit a high number of missing edges.

---
**Algorithm 1** Training Algorithm of SmGNN.

---
**Require:** $\mathcal{G} = (\mathcal{V}, \mathcal{E}_1, \ldots, \mathcal{E}_L, \mathbf{X})$
**Ensure:** GNN model with a set of parameters $\Theta$.
 1: Randomly initialize the model parameters.
 2: **repeat**
 3:     Learn relevant representation from the Eq. (3).
 4:     Guide the relevant representation to learn structure information for the sparse layer from the Eq. (6).
 5:     Augment the sparse layer's adjacency matrix from Eq. (10).
 6:     Augment the supervised information to learn expressive representation for exploring missing links for the sparse layer from the Eq. (11).
 7:     Obtain the final loss $\mathcal{L}$ from the Eq. (12).
 8:     Update $\Theta$ by minimizing $\mathcal{L}$.
 9: **until** convergence
10: **return** GNN models with parameters $\Theta$.

---

## B  Dataset

We put the detailed statistics of the dataset in Table 4.

## C  Hyperparameter Analysis

In this section, we further conduct hyperparameter analysis of $R$, which controls the number of edges added in the original edge set of the sparse layer for edge prediction in Eq. (11). In this experiment, $R$ is varied as $\{0, 100, 500, 1000, 2000\}$. The corresponding results are shown in the Figure 7. We can observe that large and small values also can't achieve good performance. Small values of $R$ result in limited supervised information being augmented for the sparse layer, making it challenging to train an effective model for link prediction. On the other hand, larger values of $R$ can introduce noisy edges when augmenting the original edge set of

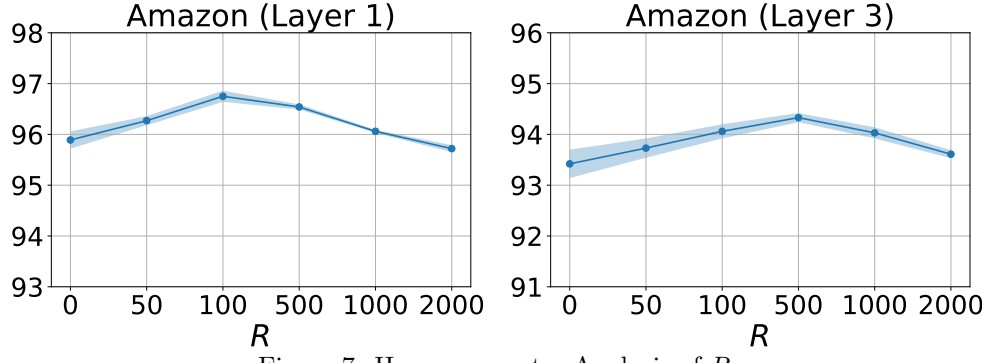

Figure 7: Hyperparameter Analysis of $R$.

the sparse layer. This incorporation of noisy edges can negatively impact the model's performance, leading to suboptimal results. From our results, a suitable range for $R$ is between 500 and 1000.

## D Time Complexity Analysis

Since our SmGNN is agnostic to GNN encoders, we consider $M$-layer GCN as an example. A $M$-layer GCN (Kipf & Welling, 2016a) with $d$ hidden dimensions has $\mathcal{O}\left(dM|\mathcal{E}| + Nd^2M\right)$ complexity, where $\mathcal{E}$ is the edge set of the graph with one single layer and $|\mathcal{E}|$ denotes the number of edges. For the GNN model in the Relevant Information Encoder module with the GNN model for the sparse layer, the time complexity is $\mathcal{O}\left(\sum_{i=1}^{L} dM|\mathcal{E}_i| + Nd^2M\right)$. Then, for the Multi-layer Encoder, the time complexity to reconstruct the adjacency matrix is $\mathcal{O}\left(\sum_{i=1}^{L} d|\mathcal{E}_i|\right)$. For the augmented graph structure, the time complexity is $\mathcal{O}\left(NKd^2\right)$. The time complexity to reconstruct the augmented edge set of the sparse layer can be $\mathcal{O}\left(d(|\mathcal{E}_L| + R)\right)$. mGCN, a multi-layer GNN model, focuses on learning multi-layer relations. In comparison, our model has additional time complexity due to the learning of augmented structures and the utilization of augmented edge sets. This extra time complexity can be represented as $\mathcal{O}\left(d(|\mathcal{E}_L| + R) + NKd^2\right)$. In our experimental setup, the number of edges in the sparse layer, denoted as $|\mathcal{E}_L|$, is typically small. Additionally, the values of $K$ and $R$ are often set to 50 and 500 respectively. As a result, our model does not incur much extra time consumption due to these factors.

## E Difference between Multi-layer Graph and Heterogeneous Graph

A multi-layer graph consists of multiple layers, each layer representing a different type of relationship or interaction between the same set of nodes. Within each layer, the nodes and edges are homogeneous, meaning they represent the same type of entities and relationships. A heterogeneous graph contains different types of nodes and edges. Although there are different types of nodes and edges, they all exist within a single, unified graph structure and there is only one edge between a pair of nodes

## F Real-world Applications of Multi-layer Graph

Many real-world graphs can be treated as multi-layer graphs, such as transportation networks, business networks, and social networks. For example, in a social network, user $u_i$ can simultaneously be a friend of $u_j$, send messages to $u_j$, and like $u_j$'s posts. Then there will be three edges between $u_i$ and $u_j$, which forms a 3-layer network. Similarly, in a business network, person $p_1$ can have a phone call with person $p_2$, send money to $p_2$, and have a meeting with $p_2$, which also forms the multi-layer network. As multi-layer graphs are pervasive, the proposed framework can have wide applications, e.g., friend suggestion by leveraging multi-types of relationships in the multi-layer social network.

