# OpenReview forum: "SmGNN: Link Prediction in Sparse Layers of Multi-layer Graphs"
_TMLR — Rejected by TMLR_

### Review · Reviewer_sF7y · 2024-02-22

**Summary Of Contributions:**

This paper deals with the problem of link prediction for multi-edge networks (there exist different types of edges). The authors assume that there is a large number of missing edges (just of one type), and they propose SmGNN, a neural network model to predict the missing edges. The model employs different GNN models for the different types of edges and uses an attention mechanism to combine the emerging node representations. A reconstructed adjacency matrix for each type of edges is produced based on those representations, while an augmented adjacency matrix is also produced for the type of missing edges and is fed to a GNN to produce final node representations and predict the adjacency matrix.

**Audience:**

Yes

**Broader Impact Concerns:**

There are no concerns.

**Claims And Evidence:**

Yes

**Requested Changes:**

- The authors need to better motivate the proposed model and its components. Please explain what benefits each component is expected to bring to the model.

- Please, perform some experiments to validate that the assumptions you make actually hold. For example, are nodes that share similar features also connected by an edge in the considered datasets?

- From the ablation study in Table 3, we can observe that even without some of its components, the proposed model would still outperform most of the baselines (or all of them). I would suggest the authors explain why does the model outperform the baselines even without those components.

- Equations 4:  $\mathbf{F}_l$ ==> $\mathbf{F}^l$

**Strengths And Weaknesses:**

Strengths:
- The proposed model shows strong empirical performance over previous baselines on the three considered datasets. In some cases, the difference in performance between the proposed SmGNN model and the baselines is significant. Thus, the model could be very useful to practitioners.

- The problem considered in the paper is properly motivated. Furthermore, this problem is interesting and deserves more exploration and attention.

- The presentation is reasonably clear.

Weaknesses:

- The main components of the SmGNN model have been utilized in other models as well. This is not a weakness according to TMLR's criteria, however, I feel that the use of these components is not properly motivated in some cases, while the model also seems unnecessarily complex. For example, it is not quite clear to me why do the authors produce the similarity matrix $\mathbf{S}^{sp}$ and why do they feed the adjacency matrix $\tilde{\mathbf{A}}^L$ to a GNN. In my view, better explanations need to be given.

- The authors make some strong assumptions which are likely to have a significant impact on the model's performance. For example, the authors assume that nodes that have similar feature vectors are more likely to be connected with each other. Even though this assumption might hold for some graphs, it might not hold for others.

- Some architectural choices do not make a lot of sense. For instance, the authors augment the set $\mathcal{E}_L^O$ (i.e., increase the number of positive samples) based on the model's predictions  ($R$ edges with highest values in $\mathbf{S}$). Using a model's predictions as labels for model training is not a standard approach and thus, the authors need to be careful.

---

> ### Author Response · Authors · 2024-03-31
> **Response to Reviewer sF7y**
>
> Thank you for your valuable review. Below we address the detailed comments and hope that you find our response satisfactory.
>
> **Q1: Motivation of each component**
>
> Thanks for your suggestion. $S$ is a similarity matrix where each element has a value between -1 and 1. If we don't set a threshold to filter out non-negative values, then the augmented adjacency matrix can contain negative numbers, and the values of the elements of the adjacency matrix should all be non-negative. Besides, it could introduce noise if we use all the values in $S$ to augment the matrix.
>
> The augmented matrix $\widetilde{\mathbf{A}}^{L}$ has the knowledge from $S$ which contains information from other layers. With the augmented graph adjacency matrix, we could learn better representations of nodes compared to Eq. (1).
>
> We have updated the Proposed Method section.
>
> **Q2: Some Assumptions**
>
> Nodes with similar feature vectors are more likely to be connected, which is a well-established concept in network analysis called graph homophily. Most of the works about GNN assume the existence of graph homophily.
>
> **Q3: Some architectural choices**
>
> Using a model's predictions as labels for model training is a technique commonly referred to as pseudo-labeling. It is a popular approach for dealing with label sparsity issues in semi-supervised learning [1,2].
>
> **Q4: Why can our model still outperform the baselines even without some components?**
>
> We can see from Table 3 that only augmenting the graph structure (w/o augment (set)) with relevant representation from other layers or only adaptively augmenting the edge set (w/o augment (adj)) can still achieve good results. The possible reason is that both components use the information from other layers and both of them can make good use of the information from the other layers, so even removing one of them can still achieve good performance.
>
> **Q5: Typos**
>
> Thanks for your careful reading. We've made the changes in our version.
>
> **Thanks again for your valuable comments. We hope our responses answer your questions and address your concerns.**
>
> ---
> [1] "Transductive semi-supervised deep learning using min-max features." ECCV. 2018.
>
> [2] "Label propagation for deep semi-supervised learning." CVPR. 2019.

---

### Review · Reviewer_Jg35 · 2024-03-11

**Summary Of Contributions:**

The authors aim to tackle the problem of sparsity in a particular layer of multiple-layer graphs (where multiple types of edges exist). Specifically, the authors propose to utilize the information from other dense layers to effectively represent nodes in the graph, and use them for predicting missing links on the sparse layer. In addition, the authors propose strategies (such as sparse layer augmentation and reconstruction) to enhance the link prediction capability for the sparse layer. The authors validate the proposed method, namely SmGNN, on three benchmark datasets, showing its effectiveness over several baselines.

**Audience:**

Yes

**Broader Impact Concerns:**

The authors do not discuss any concerns on the ethical implications of their work.

**Claims And Evidence:**

No

**Requested Changes:**

Please see the weaknesses above.

**Strengths And Weaknesses:**

### Strengths
* The link prediction problem on multi-relational graphs is less explored.
* The idea of using the information from dense layers to predict links on the other sparse layer is interesting and convincing.
* This paper is well-written and easy to follow.

---

### Weaknesses
* The assumption that considering only one sparse layer is too restrictive. There could be multiple sparse layers (in real-world applications); meanwhile, the authors focus on the scenario where only one layer is sparse and all the other layers are dense.
* The datasets used for validating the proposed SmGNN are not convincing enough. Specifically, Amazon and IMDB datasets have only three and two relations, respectively, and considering only the two or one layer as the dense layer to predict links on the other sparse layer is too restrictive. The authors may consider existing knowledge graphs as the benchmark datasets, which have various edge types with much diverse sparsities.
* The authors may tone down the claim on the novelty of introducing a link prediction problem for sparse layers on multi-layer graphs, considering the similar work [A] that presents a problem of few-shot link prediction (sparse link prediction) on multi-relational graphs.

---

### Questions
* Have you considered applying the proposed strategy on both the sparse and dense layers jointly? The dense layers could also benefit from the proposed method.

---

[A] Learning to Extrapolate Knowledge: Transductive Few-shot Out-of-Graph Link Prediction, NeurIPS 2020.

---

> ### Author Response · Authors · 2024-03-31
> **Response to Reviewer Jg35**
>
> Thank you for your careful and valuable comments. We hope our responses will help address your concerns.
>
> **Q1: What if there are multiple sparse layers and other layers containing sparse layers**
>
> There may be more than one sparse layer in real-world applications. Our research assumes that we need to perform link prediction on a certain layer, but it lacks sufficient information due to sparsity. In this case, we can leverage the information from other layers to enhance the structure of this layer, thereby enabling link prediction. Moreover, not all other layers necessarily need to be dense. Although some layers among them might be sparse, overall, the useful information is still abundant.
>
> **Q2: Datasets problems**
>
> The datasets we use are commonly used in multi-layer graphs, whereas the knowledge graph belongs to heterogeneous graphs, which is a different type of graph. A multi-layer graph consists of multiple layers, each layer representing a different type of relationship or interaction between the same set of nodes. Within each layer, the nodes and edges are homogeneous, meaning they represent the same type of entities and relationships. A heterogeneous graph contains different types of nodes and edges. Although there are different types of nodes and edges, they all exist within a single, unified graph structure and there is only one edge between a pair of nodes. If we convert a KG into a multilayer graph with each layer having one relationship, then in each layer, there will be many small disconnected subgraphs, which is not applicable.
>
> **Q3: Difference with another work**
>
> Dear review Jg35, the paper you mentioned is totally different from ours and is focusing on a different task. They focus on multi-relational graphs. The definition of a multi-relation graph is: “Let $\mathcal{E}$ and $\mathcal{R}$ be two sets of entities and relations respectively. Then a link is defined as a triplet $(e_h, r ,e_t)$, where $e_h, e_t \in \mathcal{E}$ are the head and the tail entity, and $r \in \mathcal{R}$ is a Specific type of relation between the head and tail entities. A multi-relational graph $\mathcal{G}$ is represented as a collection of triplets. That is denoted as follows: $\mathcal{G} = \{ (e_h, r ,e_t) \} \subseteq \mathcal{E} × \mathcal{R} × \mathcal{E}$. ”[1]
>
> They assume that there will be more nodes (entities) over time. Newly emerged entities often have few links. They focus on another problem which is how to predict the links between the seen and newly emerged nodes and between the newly emerged nodes. They tackle this problem with a transductive meta-learning framework. Our method uses the information of other layers to help perform the link prediction task on the sparse layer.
>
> **Q4: Applying the proposed strategy on both sparse and dense layers jointly**
>
> Dear reviewer, do you mean to apply our method on a dense layer?
>
>
> **Thanks again for your valuable comments. We hope our responses answer your questions and address your concerns.**
>
> -------------------------
>
> [1] Learning to Extrapolate Knowledge: Transductive Few-shot Out-of-Graph Link Prediction, NeurIPS 2020.

---

> > ### Comment · Reviewer_Jg35 · 2024-04-07
> >
> > Thank you for your response. My major concerns about the restrictive setting (that the authors consider only one sparse layer) and the datasets (which have only a few relations not sufficient to validate the proposed approach) still hold. For Q4, yes; I am curious what if the proposed approach is applied to a dense layer.

---

### Review · Reviewer_EmM5 · 2024-03-19

**Summary Of Contributions:**

The paper presents a novel framework, Sparse Layer Reconstruction Multi-layer Graph Neural Network (SmGNN), designed to improve link prediction in multi-layer graphs, particularly in layers that are sparse due to missing edges. SmGNN leverages information from denser layers to infer missing links in sparse layers. It uses a relevant information encoder module to fuse information from various layers and employs supervised edge prediction to refine the learning process. By augmenting the sparse layer’s graph structure with additional edges derived from other layers’ representations, SmGNN enhances the exploration of relational patterns and connections, leading to better performance in link prediction tasks.

**Audience:**

Yes

**Claims And Evidence:**

Yes

**Requested Changes:**

Please kindly refer to the questions from Strengths and Weaknesses.

**Strengths And Weaknesses:**

Strengths:
* SmGNN introduces a unique method to address the challenge of link prediction in sparse layers by leveraging information from other layers, a problem not extensively explored before.
* The framework’s encoder module intelligently fuses information from different layers, ensuring that the characteristics of the sparse layer are captured effectively.
* By augmenting the graph structure of the sparse layer with additional edges, SmGNN significantly improves the representation learning process, allowing for a more comprehensive understanding of the sparse layer’s relational patterns.

Weaknesses:
* The research scope of the paper is rather narrowed. The multi-layer graph is a subcategory of the heterogeneous graph. Compared to the fruitful research on the heterogenous graphs, the research progress of GNNs especially targeting the multi-layer graphs is relatively fewer. The authors are suggested to conduct in-depth discussions on the scenarios of practical employment of SmGNN to demonstrate its industrial scope.
* The performance of SmGNN may heavily rely on the quality and relevance of the information from other layers. If these layers contain noisy or irrelevant data, it could negatively impact the model’s ability to accurately predict links in the sparse layer. This is relatively similar to the cold-start scenario in recommendations, therefore discussions on this potential application of SmGNN are suggested to be discussed.
* While SmGNN shows promise in the experimental evaluations presented, its ability to generalize across a wide range of real-world graphs (such as KGs) with varying characteristics and degrees of sparsity remains to be thoroughly tested.
* While the SmGNN does not incur much extra time consumption, the scalability of SmGNN could be further demonstrated by experiments on larger multi-layer graphs with millions of nodes.

---

> ### Author Response · Authors · 2024-03-31
> **Response to Reviewer EmM5**
>
> Thank you for the valuable review. We appreciate the comments and would like to provide the following clarifications.
>
> **Q1: Research scope of the paper**
>
> We would like to clarify that multi-layer graphs and heterogeneous graphs are two different concepts, each with its unique characteristics and applications.
>
> A multi-layer graph consists of multiple layers, each layer representing a different type of relationship or interaction between the same set of nodes. Within each layer, the nodes and edges are homogeneous, meaning they represent the same type of entities and relationships.
>
> A heterogeneous graph contains different types of nodes and edges. Although there are different types of nodes and edges, they all exist within a single, unified graph structure and there is only one edge between a pair of nodes.
>
> Taking the transportation between cities in a state as an example, two cities may be connected by three modes of transportation: rail, road, and air. If there is both a railway and a highway between two cities, then there exists two different types of edges simultaneously between these two nodes, and the node type is only city. In this case, the transportation network can be naturally represented as a 3-layer graph. However, in a heterogeneous graph, although there may be different types of edges between nodes, there can only be one edge between two nodes.
>
> Many real-world graphs can be treated as multi-layer graphs, such as transportation networks, business networks, and social networks. For example, in a social network, user $u_i$ can simultaneously be a friend of $u_j$, send messages to $u_j$, and like $u_j$’s posts. Then there will be three edges between $u_i$ and $u_j$, which forms a 3-layer network. Similarly, in a business network, person $p_1$ can have a phone call with person $p_2$, send money to $p_2$, and have a meeting with $p_2$, which also forms the multi-layer network. As multi-layer graphs are pervasive, the proposed framework can have wide applications, e.g., friend suggestion by leveraging multi-types of relationships in the multi-layer social network.
>
> We will incorporate the discussion of the difference between heterogeneous graphs and multilayer graphs, and some real-world examples and applications of multilayer graphs into our paper. We have added two sections in Appendix in our revision.
>
> **Q2: What if other layers contain noisy or irrelevant data?**
>
> We agree that relevant information is very important, and noisy or irrelevant data could bring negative effects. However, we have proposed the Relevant Information Encoded in Section 4.1 to mitigate the issue:, “As not all layers are that highly correlated with layer $L$, those not highly relevant information might introduce noise. Hence, it is crucial to effectively gather and utilize the relevant information from some dense layers to predict the links within the sparse layer. To achieve this goal, we propose a relevant information encoder to learn the relevant node representation matrix $H$, which captures the informative aspects present in the dense layers, enabling the reconstruction of missing links within the sparse layer.” Our designed Relevant Information Encoder can help filter out the noisy or irrelevant data from other layers.
>
> **Q3: Generalize across a wide range of real-world graphs (such as KGs)**
>
> We would like to kindly remind the reviewer that all the datasets we use in the paper are real-world multi-layer graphs like Amazon and Epinions. Knowledge Graphs are heterogeneous graphs, which have at most one edge between a pair of nodes. If we convert a KG into a multilayer graph with each layer having one relationship, then in each layer, there will be many small disconnected subgraphs, which is not applicable.
>
> **Q4: Scalability of SmGNN**
>
> As we can see from Table 4, the Epinions dataset has more than three hundred thousand nodes. It is the largest multi-layer graph dataset we could find. We also have a time complexity analysis in Appendix D, which shows the scalability of SmGNN.
>
> **Thanks again for your valuable comments. We hope our responses answer your questions and address your concerns.**

---

### Decision · Action_Editor_Dbgk · 2024-04-21

**Recommendation:** Reject

**Comment:**

I'm recommending rejection as the evidence for the claims are not very solid and the scope and therefore audience is quite limited.

**Audience:**

Missing link prediction is a useful task, but the specific task considered in this paper, that assumes all layers except one are dense and the task is to predict missing links in that one layer, is a bit narrow. This significantly limits the amount of audience this paper may have.

The proposed model, even though it is new, is a simple assembly of existing components. The lack of excitement from the reviewers also indicates the potential lack of audience.

**Claims And Evidence:**

This paper claims 3 contributions:
1. They claim the problem of predicting missing links for a specific relation on multi-relational graphs is new and they investigated it.
2. They proposed a new model SmGNN for this specific problem.
3. They verified the performance of this model on a few datasets that demonstrated the effectiveness of SmGNN.

Regarding claim 1, it is hard to say the problem of missing link prediction in multi-relational graphs is a totally new problem. It is possible the very specific setting considered in this paper, assuming all but one type of relation are dense, and only predicting the missing links for one relation is new, but the scope of this is quite limited.

The proposed model SmGNN is new, but all the reviewers have raised questions about the design and the general applicability of this model.

The performance of this model on two datasets does show it performs better than alternatives. However, the evidence was questioned by reviewers, as predicting one type of relation given the other and testing on small graphs may not be convincing enough.